# Is There a Homos in Eros: Sexual Incarnation in Emmanuel Falque

**Justin Leavitt Pearl**

The Atkins Center for Ethics, Carnegie Mellon University, Pittsburgh, PA 15213, USA; jpearl@andrew.cmu.edu

**Abstract:** Since the fifth of Edmund Husserl's *Cartesian Meditations*, the encounter with the Other has been a central locus of phenomenological research. This centrality is nowhere as clear as in the phenomenological study of gender and sexuality. Just as the erasure of alterity is understood to render ethics impossible, so too is the erasure of sexual difference taken to render genuine erotic love impossible. Employing the recent work of Emmanuel Falque, this investigation aims to interrogate the way in which this rhetoric of sexual alterity has often served to maintain and reinforce a logic of homophobia and queer erasure. If every major phenomenologist of the past century has agreed with Emmanuel Falque, that there is a "*heteros* in *eros*, or an other in difference", I will here ask the question: might there not also be a *homos* in *eros*?

**Keywords:** Emmanuel Falque; phenomenology; sexuality; queer theory; alterity

## 1. Difference and Alterity

Since the fifth of Edmund Husserl's *Cartesian Meditations* ([Husserl 1977](#)), the encounter with the other has been a central locus of phenomenological research. In subsequent years, thanks in no small part to the philosophical and ethical investigations of Emmanuel Levinas, this focus has only intensified, and the maintenance of alterity has continued to be an overriding concern and chief ethical value among phenomenologists.

This centrality is nowhere as clear as in the phenomenological study of gender and sexuality. Already in *Time and the Other*, Levinas could describe the feminine as "the absolutely contrary contrary . . . absolutely other" ([Levinas 1987](#), p. 85). In this way, Levinas solidified sexual difference as the central concern of phenomenological analyses of erotic love and gender—sexuality is the encounter with alterity par excellence.[1] Feminist critiques of the seeming asymmetry of this alterity emerged almost immediately. In *The Second Sex*, Simone de Beauvoir takes Levinas to task for his androcentrism, writing that, according to Western philosophical canon up through Levinas, "he is the Subject, he is the Absolute—she is the Other" ([de Beauvoir 1974](#), p. xxii). In a similar manner, Luce Irigaray would later parody Levinas, in "The Fecundity of the Caress", writing: "beloved woman. Not female lover. Necessarily an object, not a subject with a relation, like his, to time" ([Irigaray 2001](#), p. 126).[2]

Nevertheless, despite these feminist critiques, erotic love continues to be read primarily through the lens of alterity and sexual difference—which even now serves as the unquestioned centerpiece of phenomenological reflections on sexuality. Just as the erasure of alterity is understood to render ethics impossible, so too is the erasure of sexual difference taken to render genuine eros impossible.

In broad terms, this thesis is basically correct. Eros is deeply and irreducibly—according to Husserl, essentially ([Husserl 1981](#), p. 335–37)—intersubjective. Eros is about the encounter with the Other in their alterity. Thus, the aim of the present investigation is *not* to substitute a notion of ontological or phenomenological sameness or identity for sexual difference. Nevertheless, I will here aim to interrogate the way in which this rhetoric of sexual alterity has served to maintain and reinforce a logic of homophobia and queer

erasure. In the end, I will suggest that, quite against accusations that queer sexuality erases sexual difference, it embodies the profound diversity of erotic manifestation. In its refusal of normative heterosexuality, queer eros carries sexual alterity within itself. Thus, if every major phenomenologist of the past century has agreed with Emmanuel Falque, that there is a "*heteros* in *eros*, or an other in difference" (Falque 2016b, p. xxiv), might we not also suggest, albeit playfully, that there is also a *homos* in *eros*?

I will employ the recent work of Emmanuel Falque as a case study in this investigation not because his account of sexual difference is particularly egregious. In fact, you can find much more violently homophobic rhetoric in the early Levinasian inflected philosophy of Enrique Dussel (discussed below) or in the work of Jean-Luc Marion. Rather, Emmanuel Falque is useful precisely because his erasure of queer sexuality comes amidst a careful and nuanced phenomenological study of sexuality that is, by all appearances, well intentioned and measured. In fact, unlike his contemporary, Marion, Falque largely avoids outright homophobic rhetoric in his writings and public speeches. In this way, his queer erasure is more subtle and nuanced, and therefore, I would suggest, more generative to engage with and more important to unpack.

### 2. Eros and Incarnation

For Emmanuel Falque, eros stands at the center of proper theological and philosophical investigation, because sexuality—together with birth and death—are central figures of a finitude that is fundamentally constitutive of human life.[3] Here, the deep debt to Martin Heidegger's existential phenomenology is particularly evident. Yet, whereas Heidegger's ontological and linguistic approach turned him away from the body and sexuality (as Sartre notes, *Dasein* "appears to us as asexual" (Sartre 1956, p. 498) (On this point see: Aho (2009))), Falque's preoccupation with the material, organic, and fleshly experience of lived reality pushes the question of sexual embodiment to the center of his phenomenological project. This shift is compounded by the significant theological influence of Hans Urs von Balthasar, who's nuptial focus on sexual difference as an ontological principle likewise forms a key predecessor to Falque's theological investigations.

Thus, in a manner properly befitting Falque's attempt to draw together philosophy *and* theology, sexuality becomes a central figure of both theological *and* philosophical import, the place where an all-too-human eros and a divine agape are drawn into a tangled relation.[4] This tangled relation between human eros and divine agape reaches its apogee, its quite literal apotheosis, in the incarnation and the eucharist, where Falque detects a model for thinking about human sexuality.[5] That is to say, for Falque, particularly in his philosophical Triduum, *The Guide to Gethsemane*, *The Metamorphosis of Finitude*, and (most importantly) *The Wedding Feast of the Lamb*, the incarnation becomes the model by which human finitude—and particularly, in the present context, sexuality—should be understood. As he writes, the incarnate Christ "takes on fully the corporal modality of the present, or the gift of the body . . . in the unique, almost conjugal, moment of the act of love, in which his body is given to the other: *Hoc est enim corpus meum*" (Falque 2012, p. 3).

This eucharistic language of the giving of one's body to the other—"here is my body, given for you"—is central to the meaning of love and sexuality for Falque. Here, Falque echoes the tradition which centers the notion of "the gift", theologically in von Balthasar and phenomenologically in Jean-Luc Marion.[6] "The first and most ardent 'desire' of God, as also of humankind", Falque can say, speaking in one voice of both sexuality and the eucharist, "is, paradoxically, to meet a body and to be given bodily" (Falque 2016b, p. 154). Yet, in both instances—sexuality and the eucharist—the nature of the gift can only be understood in relation to the recipient, and this recipient is, ontologically speaking, the same in both instances. The recipient is *the Other*.

This alterity is, for Falque (following Marion), a phenomenological requirement. For the gift to be truly given, for it not to fall back into the economy of exchange, the gift must be given to that one which is, as Marion writes, "by definition transcendent because Other" (Marion 2002b, p. 88). In the case of sexuality, therefore, this necessitates the gift of

one's body to the *sexual Other*. On this ground sexual difference becomes a pivotal focus of Falque's analysis of eros. To again cite his central principle: "There is *heteros* in *eros*, or an other in difference".

## 3. Sexual Difference

In *The Wedding Feast of the Lamb*, Falque paraphrases Luce Irigaray's famous axiom, "Sexual difference is probably the issue of our time, which could be our 'salvation' if we thought it through" (Falque 2016b, p. 138).[7] The primacy given to the notion of sexual difference in both his philosophical and theological reflections suggest the seriousness with which he takes this axiom. Yet, he is by no means the first phenomenologist to tackle the problem of sexual difference. Falque remains, for example, justly in my opinion, deeply critical of Levinas' account of the feminine. As Faqlue writes:

> Levinas certainly proposes a philosophical treatment of femininity as a site of interiority, or a topos where woman's 'mode of being . . . consists in slipping away from, the light.' We cannot, however, seriously maintain this—at least, not without deceiving ourselves. To hold that femininity is interior, and masculinity exterior, is not adequate as a way of marking a difference that is, to say the least, constitutive and originary. It does not work in a humanity that, at least today, will not stand being divided up in such a way. (Falque 2016b, p. 138)

Here, Falque does two things. First, he alludes to (and implicitly affirms) the feminist critiques of Levinas, already apparent in de Beauvoir and Irigaray. Whatever sexual difference is for Falque, it will not be reducible to the cultural tropes of masculinity and femininity, nor to a mythological notion of woman as the mysterious dark continent. Likewise, and despite his interest in reaffirming the organicity of the human body, Falque will likewise deny that sexual difference can be reduced to genital differentiation—it is, he often insists, the man–woman, and not the male–female difference that interests him, that he takes to have an ontological import.[8]

In this way, Falque situates the question of sexual difference as "originary", that is to say, ontologically constitutive. "The erotic difference, or, in other words, the sexual difference", he writes, "is there from the start" (Falque 2016b, p. 137). Here, Falque's comparison of two myths of the creation of sexual difference is useful. In Aristophanes' account of sexual differentiation in Plato's *Symposium* (189d–193d), sexual difference is a punishment meted out by Zeus in response to human hubris.[9] Humanity is thereby torn in two. Eros fulfills the task of "looking for our other half" (192d), in order to restore the original goodness of unity. Falque contrasts this Platonic myth with the account of sexual differentiation found in Genesis 2. There, he writes:

> To separate the man from the woman here is not to punish them but, on the contrary, to build them, to create through the act of differentiating them. The difference is good in Christianity—whether it is a question of that between uncreated and created or of that between man and woman, the difference at the heart of humanity. (Falque 2012, p. 99)

Where Aristophanes envisioned division as a problem to be overcome in the search for an original unity, Falque finds in the Jewish creation account of Genesis 2, and more particularly in its interpretation by Christ in the Christian Gospels,[10] an affirmation of the creative fecundity of difference: "in the eyes of the founder of Christianity, there [was] no other origin but sexual differentiation, or no other beginning but humanity differentiated" (Falque 2016b, p. 140). "We can", he writes elsewhere, therefore "set out in praise of difference, to show that nothing is to be feared more than uniformity, whether we are speaking of reducing everything to sameness, or of making all identical" (Falque 2016b, p. 60).

According to Falque, this originary ontological difference is iterated across multiple levels. It manifests within the Godhead, where the trinity emerges from the differentiating love between the divine persons.[11] It manifests in the ontological difference between creator

and creation. It cuts across humanity, dividing it in two—man and woman. Each of these levels are willed by God,[12] who, on Falque's account, glorifies limitation and differentiation: "God is (the) difference from which all differences come" (Falque 2016b, p. 153). These originary differences, particularly the originary difference between man and woman, serve two roles in Falque's thought: they are revelatory of identity and productive of love.

On this first point, Falque remains harshly critical of any attempt to posit one's own identity. "Difference" he suggests, "produces the most radical strangeness but also gives birth to the self and its true identity" (Falque 2016b, p. 168). Falque is doing two things here. First, at a social and political level, and in a point that will be discussed more fully below, he is critiquing contemporary accounts of gender constitution, which he takes (wrongly, I would suggest) to argue for the individual (or individualistic) constitution of the self and its identity. Second, at a phenomenological level, he is following recent moves in French phenomenology which have sought to rethink the self as constituted externally by a transcendent Other—or, in the grammatical language often here employed, to shift from thinking the self in the nominative, to thinking of it in the accusative (Levinas) or the dative (Marion). Our identity is not something we originally have or are, but something we are always first given by the Other—whether the divine Other or the sexual Other. As he writes, "humankind discovers difference then, in the otherness of masculine and feminine that constitutes the one and the other" (Falque 2016b, p. 65).[13]

This revelation of the self, given by the Other, appears most directly in the sexual encounter—here rendered nuptial, insofar as Falque (somewhat concerningly) uses the pairs man/woman and husband/wife interchangeably.[14] There, in the nuptial "night of love", I am paradoxically given to myself by the other at precisely the point that the other gives their body to me ("Here is my body, given for you"). Thus, the masculinity of the man is found in the femininity of his partner, and the femininity of the woman is found in the masculinity of her partner. As Falque writes:

> The husband's experience of his wife is experience as a man and the wife's experience of her husband is experience as a woman. To make one flesh is not to renounce one's own flesh—far from it. The flesh of the other sends me back to my own flesh, as my flesh in relation to the other's flesh, such that we can never truly dissolve ourselves in a unity of the flesh. (Falque 2016b, p. 61)

Or again:

> Man is never so much masculinized as when he encounters his woman (wife) erotically, and woman never so much feminized as when she is united in terms of the flesh with her man (husband). . . . The union of flesh . . . intensifies the difference and recognizes it also as something to be lived. (Falque 2016b, p. 153)

This constitution by the other is not only generative of the self in its ontological reality (as *Dasein*), but, as Falque is a thinker of the organic, is also generative of our concrete embodiment: "one becomes one's own body in uniting with the body of the other (human in difference from God, masculine or feminine in difference from each other)" (Falque 2016b, p. 158).

Having constituted two selves—man and woman, husband and wife—sexual difference also serves to constitute the love that arises between them. "The 'man-woman' difference", Falque argues, "is the primordial stuff and the difference of everything, in which God has designed all differentiation as well as all acts of love" (Falque 2016b, p. 141). For Falque, despite the language of "one flesh" found in Genesis 2, love should not be misunderstood as the rejection of difference in favor of a primordial unity (what Falque takes as Aristophanes' approach), but rather in the embrace and valorization of sexual difference itself. Indeed, he argues, "man and woman are marked irremediably here by a lack, or perhaps we should say, by a difference. It is always in being different, one from the other, and in differentiating themselves, that the couple are able to love" (Falque 2016b, p. xxi).

Here, unfortunately, in what is a recurring feature of Falque's analysis of difference, little is said about the mechanism at play. While he goes to great lengths to show why the rejection of difference is incompatible with the posture of genuine love, he fails to explain the way in which sexual difference is positively generative of love. Thus, he writes, for example, that, "there are aspects of man that 'are not' of women, and aspects of women that 'are not' of men, since to love within a shared humanity is not to renounce differentiation" (Falque 2016b, p. 61). And yet, he fails to mark what these uniquely gendered "aspects" men and women are, or how their difference is generative of love. Falque insistently affirms a binary man/woman gender differentiation, but having rejected biological, mythological, and cultural markers of this difference, it remains fundamentally unclear what constitutes sexual difference in his eyes. This lack of clarity produces serious difficulties in his account of eros.

## 4. Limit and the Law of Nature

Rejecting the sharp methodological delimitation of philosophy and theology often insisted upon by his contemporaries (Marion 1994), Falque seamlessly slips between theological and philosophical registers. In his preferred terminology, he is interested in "crossing the rubicon" between Rome (theology) and Paris (philosophy).[15] This cross-disciplinarity is exemplified by his paired principles: "the more we do philosophy, the better the theology" (Falque 2012, p. x); and "the more we theologize, the better we philosophize" (Falque 2015, p. 16). This disciplinary flexibility is in some tension Falque's marked preference for clear limits within his broader philosophical and theological investigations. Thus, while he will sometimes describe his project as the "transgression of boundaries" (Falque 2015, p. 5), he insists that his aim is to take philosophy "to its limit", but *not over* its limit—he will theologize and philosophize "without transgressing the fixed bounds of rationality" (Falque 2016b, p. 11). As he insists, "One enters the other's field in order to respect the boundaries" (Falque 2016a, p. 138).

Falque's concern for limit emerges directly out of the primacy of difference. For him, limits are generative of difference: "Without limits there is no differentiation, only the myth of fusion or the trap of a false union" (Falque 2016b, p. 154). In particular, Falque identifies the productivity of those limits which mark the difference between creator and creation, and between man and woman. In both instances, his preferred language to think these limits is "nature". In this way there is, in Falque's theology, something of an appeal to natural law.

These limits are natural, according to Falque, because they emerge from God—the Unlimited who nonetheless loves the limit. To the question "why did God create humans?", for example, Falque suggests that there is only one answer: "because of his love and respect for limits, he the Unlimited had need of a different limited one, capable of being itself for him as a face-to-face for his own act of loving" (Falque 2016b, p. 148). That is to say, because love requires difference and difference requires limit; God necessarily requires a delimited being (i.e., a finite being) in order to fully express love. Here, in Falque's own words: "phenomenological finitude is reread theologically as a kind of theological limit wished for and desired by God" (Falque 2016b, p. 152). In this passing statement, Falque summarizes the center of his ethical and practical philosophy. In Falque, the descriptive truths of existential phenomenology are theologically transfigured into the normative truths of natural law. As he writes, "'finitude' in phenomenology (Husserl) joins here with 'limits' in theology (Aquinas)" (Falque 2016b, p. 224). Thus, for example, we should not, on Falque's account, "wrongly [regret] that we are not like [God], not infinite" (Falque 2016b, p. 152). To do so would be to reject the "natural goodness" of limit. Likewise, when he critiques "angelism"—a posture that rejects the goodness of fleshly embodiment—he presents it as "a way of going beyond frontiers" (Falque 2016b, p. 72). "God does not call us to angelism", he argues, "as if we were illicitly to go beyond the limits of our created being" (Falque 2016b, p. 52). This legal language of "illicit" is key here. Falque is making a moral argument on the grounds of natural law.

Thus, if there is an original sin for Falque, then its form is—perhaps ironically, given his own methodological fluidity—the illicit crossing of boundaries. Indeed, he will fundamentally redefine sin itself—or, as he prefers to name it, "bestiality" (which he holds in opposition to the goodness of "animality")—as the illicit crossing of boundaries. Humans, he argues, "can fall not into animality of which they are already constituted, but into bestiality, where it is uniquely possible to veer off course from nature itself, and to founder below one's own nature" (Falque 2016b, p. 73).

This ontological limit between creature and creator, between human and God, is unsurprisingly taken as an unproblematic justification for the imposition of subsequent limits *within* the human world. Particularly, Falque is invested in the transgression of limits between man and woman, and between licit and illicit manifestations of sexuality. As he writes:

> The difference from the divine is given fully to the human being, as the difference of the masculine is given to the feminine and becomes visible there. The man (husband) is all the more man in that he receives his masculinity and its difference in giving himself to his woman (wife)—like God now, who is, at least for us, all the more God because he reveals himself in his divinity in being given to our humanity. (Falque 2016b, p. 157)

Thus, when Falque writes of a "metaphysical basis of the man-woman difference" (Falque 2016b, p. 61), when he writes that "God created us in 'difference'—men (*ish*) and women (*ishah*)—and wished that we should never cease to 'differ'", (Falque 2016b, p. xxiii) this can be understood as based in a normative conception of human nature. "Sexual difference" he writes, is "not simply a cultural matter (gender theory) but also a matter of nature (taking a new and necessary look at the basic concepts of natural theology)" (Falque 2016b, p. 61). Or again, he writes, explicitly:

> Far from being cultural (as in gender theory), the difference of the sexes is given to us first of all as natural and remains something that we cannot shrug off. But our nature is not based on genital difference except insofar as it is also sexualized. . . . There is no sexual difference beyond a nature onto which desire is grafted and that modifies it. Without this, the genital (male-female difference) would never pass into sexuality (man-woman difference). (Falque 2016b, p. 144)

All of this to say, sexual difference, for Falque, is not cultural; it is natural—but understood not in the contemporary sense of biological (e.g., in the natural sciences), but in the classical sense as essential or ontological.

Nevertheless, Falque's positive account of this ontological difference remains ambiguous. Once again, having rejected cultural, mythical, and biological determinations of sexual difference, it is fundamentally unclear where Falque roots sexual difference. However, what is clear, is that he fixes this difference directly to erotic desire—here understood through an explicitly heterosexual matrix. Thus, for him, the nuptial pair, husband and wife, because of their binary alterity, become the paradigmatic if not essential figure in the constitution of genuine eros. As he writes, "it is always in being different, one from the other, and in differentiating themselves, that *the couple are able to love*" (Falque 2016b, p. xxi). This is a telling formulation. If it is only by their sexual difference that the nuptial couple is able to love, then genuine eros appears impossible outside of these specific constraints.

The socially and politically fraught nature of this construction of sexual difference is clearly seen in Falque's occasional drift toward patriarchal figures of the masculine. Falque writes, for example, "the world is a boundary limit for God, woman is a limit for man, and the animal is a limit for both humankind and God" (Falque 2016b, pp. xxi–xxiii). What is telling in this series of limits is the implied associations. The world is to God, as woman is to man, as animal is to humankind and God. In this way the masculine is directly associated with the divine (in the first relation) and with both God and humankind understood in its universal essence (in the third relation). Whereas woman is understood in relation to the world (in the first relation) and the animal (in the third). Additionally, while Falque's

phenomenological and theological reflections are built around the importance of terms like world and animality, which for him stand as key features of human finitude, not to be forgotten or excised, one should not fail to recognize the way in which his analogies recapitulate the most traditional denigration of women as material, fleshly, and corporeal, while at the same time elevating the masculine as universal, spiritual, and transcendent. If he *explicitly* denies Levinas' relegation of women to the sphere of immanence and the home, he nevertheless *implicitly* reaffirms it. Consider the following interpretation of the side panels of The Ghent Altarpiece.

> Woman and man—man and woman: the difference here is not simply that of the human and the divine (Mary and John the Baptist are both shown as belonging to humanity); it is also of femininity and masculinity. Mary, turned toward her inner self (and there is so much within that inner self), is shown reading a book, mouth slightly open as if ready to take communion. John the Baptist, looking outward (and so much is in the outside world), points with his finger announcing the Lamb of God figured in the altarpiece. . . . As Aristotle tells us, the female 'is that which generates in itself' (immanence), and the male is 'that which generates in another'. (Falque 2016b, p. xxiv)

Here, in pursuit of a binary sexual difference, Falque has stumbled into exactly that reductive conception of the feminine that he had earlier rejected in Levinas.

Even more than in his occasional recourse to gendered tropes of femininity, the failure of this model of binary sexuality manifests in two related aspects of Falque's account of eros. First, in his attempt to delimit the specific forms of sexuality that are illicit, that is, his theory of perversion. Second, in his constitution of a phenomenology of eros and sexual difference in which homosexuality (if not queer love as such) is not merely normatively denigrated, but indeed, *definitionally impossible*.

## 5. Perversion and Queer Erasure

In order to unpack the transgression of proper sexual boundaries, Falque will, like Henry and Marion before him (Marion 2007, p. 165; Henry 2015, pp. 219–20), generate a fairly generic list of sexual offenses which "pervert" the natural goodness of our animality: including "prostitution, pornography, sadomasochism, [and] initiation rites" (Falque 2016b, p. 70), in one instance, and "pornography, prostitution, [and] perversion of the self" (Falque 2016b, p. 75) (this third is unclear and may refer to masturbation) in another. These are, for Falque, "a consummation 'of bodies' without loving hearts in the erotic act" (Falque 2016b, p. 137). While Falque does not explicitly explain why these specific forms of sexuality are illicit, rather than others, context may be helpful here. For Falque, as already noted above, the essential feature of sin (or, in his preferred langauge, bestiality) is the illicit crossing of boundaries. In each of these instances, an argument may be made (though Falque does not explicitly make it) that a boundary line has been crossed. Helpfully, Jean-Luc Marion, Falques *Habilitation* director, provides precisely this missing argument in *The Erotic Phenomenon*, released only a few years before Falque's own analysis in *The Wedding Feast of the Lamb*. There, Marion writes of sadomasochism, homosexuality, and bestiality, for example:

> I will constrain the flesh of the other as well as my own flesh at any price, through every trick and every convention: make-up, disguise, masks, play-acting. . . . it is necessary that I push bodies, and thus their naturalness, to their limits, or even beyond these limits. . . . Thus I will transgress the borderline of excitations, passing from pain to pleasure; I will transgress the borderline between the sexes; I could almost end up—why not?—transgressing the borderline between species. (Marion 2007, p. 165)

Given his even greater embrace of the logic of the "transgression of the boundary", it seems probable that this is precisely the sort of argument that Falque is assuming. For him, there is a natural, God-given determination of sexuality—that is, the binary coupling of

the monogamous cisgender heterosexual couple, husband and wife—which is exclusively normative. Perhaps to his credit, Falque does not immediately dismiss all non-nuptial sexuality out of hand (as Marion does, for example), suggesting that "the union of flesh is not 'better' in sacramental marriage; it is 'other,' or rather, differently oriented" (Falque 2016b, p. 170). Yet, one might question his consistency on this point, insofar as his ultimate analysis (and his conflation of man/woman and husband/wife) seems far less generous to such sexual diversity.

This lack of generosity can be seen with particular acuity in the case of queer sexuality. Same-sex sexuality and other forms of queer sexuality are not only textually absent from Falque's analyses of love and sexuality, but indeed their presence appears to constitute something of an ontological impossibility. This lacuna in Falque's work was first identified by Christina Gschwandtner, who remarks in a key footnote to *Postmodern Apologetics?* that, "one should probably point out in this context that Falque's treatment is decidedly heterosexual and seems to make homosexuality impossible, or at least deeply problematic. While he does not explicitly condemn homosexuality, any consideration of it is entirely excluded from his account" (Gschwandtner 2013, p. 316).[16]

On this point, Gschwandtner is unquestionably right. Though, perhaps, even a bit too generous. While it is quite true that Falque avoids any direct confrontation with queer sexuality in his work, his periodic references to "gender theory" (which the reader may have already noted above) appear to serve the role of a metonym. In Falque's work, "gender theory" serves as a substitution for queer sexual embodiment, as it does (together with similar dog-whistles, such as "the trans agenda") in popular anti-queer polemics. On his account, gender theory is a "blind alley" (Falque 2016b, p. 162), a paradigm in which the natural goodness of sexual difference is rejected in favor of a narcissistic preoccupation with sameness and self-constitution. In his interpretation of "gender theory", sexual difference is "cultural" and can be easily "shrug[ged] off" (Falque 2016b, p. 144). "*I* identify" (Falque 2016b, p. 168) my own self, rather than celebrating the difference that genuinely "differentiates me" and therefore gives me to myself. In this construction of gender theory, the "I" of the gender theorist is precisely the narcissistic nominative "I" against which French phenomenology since Levinas has rallied against.

It is therefore unsurprising that Falque insists that "there is *heteros* in *eros*, or an other in difference" (Falque 2016b, p. xxiv). This *heteros* is the sexual difference of the man-woman distinction. Additionally, by situating it as an essential feature within eros, Falque has shown his hand: in his phenomenology of eros, there is no room for queer sexuality. In fact, he directly says as much, linking the admission and valuation of sexual difference directly to heterosexual manifestations of sexuality. As he writes:

> Sexual difference in fact appears to be such—at least in heterosexuality, which *constitutes its modality*—that one could never experience, either physiologically or affectively, exactly what is felt by the other sex. The greater the difference, the greater the strangeness, but also the more alterity that remains. (Falque 2016b, p. 167)

This short passage is key. For Falque, heterosexuality constitutes the modality of sexual difference as such, because it is only there that the alterity of sexual difference is valorized.

This is not a unique argument for Falque. Rather, he is drawing on a decades-old argument, a rhetoric of difference that has—through a particular reading of Levinas—interpreted homosexuality as a narcissistic rejection of alterity. The philosopher of liberation, Enrique Dussel, for example, following an explicitly Levinasian line of argument, writes that "feminist homosexuality ends up summing up all perversions . . . [it is] a radical loss of [the] sense of the reality of the Other" (as cited in Vuola 2002, pp. 192–93; On this point, see: Althaus-Reid 2009, pp. 10–13; Althaus-Reid 2000, pp. 195–200; Zapata 1997, pp. 69–97; and Barber 1998, pp. 66–67). Additionally, while Dussel himself would eventually reject this position in the 90s, softening his stance on queer sexuality, Falque continues to maintain this argument, albeit more subtly. When he writes, for example, "it is enough to know one

another as sexually differentiated to love one another as different" ([Falque 2016b](#), p. 134), it is clear that he sees sexual difference—understood as the binary difference between man and woman in the heterosexual encounter—as an essential feature of love itself: "there is *heteros* in *eros*". The lesbian woman and the gay man are, on such an account, incapable of love. Indeed, insofar as the sexual encounter with the alterity of sexual difference is fundamental to the differentiation and identification of the self, the lesbian is no longer a woman at all; the gay man is no longer genuinely a man. Queer sexuality, in short, disrupts the ontological status of the individual in their deepest reality insofar as it renders intersubjectivity, identity, and love impossible.

And yet, what is clear is that, despite this rhetoric of difference, this explicit recourse to alterity is ultimately not about difference, but about the normative power of sameness. The ostensible appeal to the difference of sexual difference, masks a more insistent appeal to the sameness of heterosexuality and a rigid gender binary. As the Lacanian psychoanalyst Alenka Zupančič formulates this critique:

> The traditional division between masculine and feminine worlds, . . . actually does not see sexual difference as difference, but as a question of belonging to two separate worlds, which are 'different' from a neutral bird's-eye description, but otherwise coexist as integral parts in the hierarchy of a higher cosmic order, the wholeness and unity of which is in no way threatened by this 'difference.' These are parts that 'know their place.' And feminism (as a political movement) puts in question, and breaks, precisely this unity of the world. ([Zupančič 2017](#), p. 36)

That is to say, Falque's insistence on a binary sexual difference—one which forecloses queer sexuality, making queer love definitionally impossible—is, despite its contestations to the opposite, concerned not with difference, but with the repetition of the same.

Within queer theory, it is the work of Leo Bersani (particularly his 1996 *Homos*) that perhaps best exemplifies the rejection of this view. For Bersani, homosexual desire may be understood as a "desire for the same" but always "from the perspective of a self already identified as different from itself" ([Bersani 1996](#), p. 57). That is to say, queer sexuality does not reject difference and alterity, but quite to the contrary, fully inhabits it. The queer person is the one who *lives sexual difference within their own body*, insofar as they differ from the socially constructed normative constraints that Falque seeks to police.

The limitation of Falque's analysis of sexual difference, therefore, is that he seeks to situate the gaps and fissures of sexual difference *between* the genders. The truth is more complicated. It is not a cut between the sexes or genders that constitutes sexual difference, but rather a cut through or within each individual. As the cultural theorist Slavoj Žižek argues, "the lesson here is that sexual difference qua the Real of an antagonism is not the difference between the two sexes (masculine and feminine), but a difference/antagonism which runs across (traverses) each of the two sexes, introducing a gap of inconsistency into its very heart" ([Žižek 2017](#), p. 68).[17]

On this point, Žižek is glossing the work of Zupančič, who argues—together with Falque, that sexuality is organized around a lack. As Falque notes: "man and woman are marked irremediably here by a lack". However, where Falque figures this lack as the complementarian difference between the couple, "in being different, one from the other" ([Falque 2016b](#), p. xxi), Zupančič argues that this lack is immanent to each subject. As she writes, "this, for example, is precisely what the Lacanian formulas of sexuation force us to think: not the contradiction between 'opposite' sexes, but the contradiction inherent to both, 'barring' them both from within" ([Zupančič 2017](#), p. 72). Sexuality is not birthed in the encounter with the gendered Other, but in one's encounter with something other within oneself, an alterity which cannot be captured by subjectivity and domesticated into the sameness of self-identity.

Lacking this deeper account of sexuation, Falque is forced to ontologically police the maintenance of a simplified conception of sexual difference and alterity, through the rigid enforcement of the male/female, man/woman, husband/wife boundary. Any sexuality which threatens that borderline—whether the gender fluidity of contemporary

trans embodiment or the sexual orientation of same-sex desire—must be placed outside of the bounds of genuine eros in order to avoid "transgressing" the supposed purity of this line of demarcation.

The alternative to this policed boundary of sexual dimorphism is an openness to the true diversity and alterity of sexual manifestation. "There are two sexes on earth", writes the queer theorist Guy Hocquenghem, "but this is only to hide the fact that there are three, four, ten, thousands, once you throw that old hag of the idea of nature overboard" (Hocquenghem 2010, p. 69). The deepest alterity of sexual difference is not found in the rigid adherence to natural law and its binary logic of sexual difference; it is found in the explosive possibilities of concrete, lived sexual embodiment.

Here, a longer citation of Enrique Dussel may be particularly helpful. As noted above, Dussel's early work represented a particularly intense manifestation of a homophobic logic grounded on a theory of alterity. However, beginning in the 1990s, Dussel came to see the limitations of this perspective, such that by 2000 he could fully reverse his position and unambiguously reject his prior view, writing:

> I mistakenly interpreted as perverse 'the love of the Same for the Same,' homosexuality, radical feminism, and abortion as a negation of the Other (*filicide*). I did not take note that the Other (*la Otra*) is the alterity of the personhood of the Other (or *el Otro* [male Other]) in homosexuality, and not only 'the same' sex. I did not take note that radical feminist movements, which espoused lesbianism, would also organize in the South and, furthermore, that the radical feminism of the North had virtues that in the South we had yet to discover. In fact, this is what made possible the criticism of the radical feminism from the North, the love of the same by the same, together with the support for the 'liberation of women,' love of the alterity of the Other". (Dussel 2000, p. 265)

Here, Dussel offers two lessons for the theorist concerned with both maintaining the alterity of sexual difference and avoiding a rigid binary system of normative sexuality. First, Dussel performatively enacts his concern for alterity by genuinely taking on the critiques which were levelled at him by his feminist and queer critics. For him, the other is not an abstract structural form within his ontological and epistemological system, but real living individuals, with their own projects, ideas, and perspectives. By engaging with these others in good faith, he found his own conceptions of gender and sexuality genuinely transformed. Second, in this transformation of his thought, he unlocks precisely the linchpin absent from Falque's phenomenology of eros. The love of "the same for the same" is not a rejection of sexual difference, because it exists in relation to the total personhood of the sexual other. In writing, "I did not take note that the Other (*la Otra*) is the alterity of the personhood of the Other (or *el Otro* [male Other]) in homosexuality, and not only 'the same' sex", Dussel recognizes that possessing the same sex as one's partner does not preclude an embrace of erotic alterity, because, in the words of Jacques Derrida, "every other is wholly other (*tout autre est tout autre*)" (Derrida 1997, p. 232).

This approach identifies a primordial difference, one which rests below and before the social distinction between man and woman. As Bersani remarks: "My argument is not that homosexuals are better than heterosexuals. Instead, it is to suggest that same-sex desire, while it excludes the other sex as its object, presupposes a desiring subject for whom the antagonism between the different and the same no longer exists" (Bersani 1996, pp. 59–60). If eros depends upon sexual difference, then it is precisely the wild diversity of gender and sexual manifestations that enlivens eros and opens it to greater consummation. Returning to the language of Falque, we might say, to truly embrace difference and alterity, to "set out in praise of difference", to truly insist that there is a "heteros in eros", one must at the same time, counterintuitively recognize that there is likewise a "homos in eros".

**Funding:** This research received no external funding.

**Institutional Review Board Statement:** Not applicable.

**Informed Consent Statement:** Not applicable.

**Data Availability Statement:** Not applicable.

**Conflicts of Interest:** The author declares no conflict of interest.

## Notes

[1]　This view is subsequently taken up again and expanded in Levinas (1969, pp. 254–77) and Levinas (1985, pp. 63–72).

[2]　It is worth noting that some contemporary readers of Levinas have offered more generous or constructive readings of Levinas on gender, important titles include: Bergo (2018); Guenther (2006); Katz (2001); and Podolsky (2016).

[3]　According to Falque, the constitutive finitude of human life means that "our world becomes birth, sexuality, and death" (Falque 2012, p. 136).

[4]　"I would not want to suggest a complete 'univocity (Marion) between eros and agape (i.e., that the words are used with the same meaning and in the same sense), nor that there is a complete 'equivocity' (Nygren) (i.e., that the words are used with a different meaning and in different senses). I do not agree with the latter (equivocity) because it risks separating divine charity and human love to such an extent that nothing remains in common between them. On the other hand, univocity reduces the form of divine love so thoroughly to its model of human love that nothing remains in it that is specific to God" (Falque 2016b, pp. 47–48).

[5]　"The erotic is not fulfilled for a couple unless God contains and transforms them in his agape" (Falque 2012, p. 134). Or again; "To love myself in the flesh, and thus to assume it as specifically mine (*sua*) first requires the attestation that another, in his own flesh, constitutes it before I myself adopt it, or better, receive it. Only another, in his flesh and in an 'interlacing of flesh,' gives to me the world. So, in an exemplary way, we can say that nothing is given to me apart from the recognition that it is by the flesh of Christ alone that true access to the love of my own flesh is opened for me" (Falque 2015, p. 153).

[6]　This theme first emerges in Marion's "phenomenological trilogy," but consistently carried forward throughout his subsequent career. See: Marion (1998; 2002a; 2002b).

[7]　Paraphrasing: "Each age has one issue to think through, and only one. Sexual difference is probably the issue of our time, which could be our 'salvation' if we thought it through" (Irigaray 1993, p. 5).

[8]　"The difference man-woman (*ish-ishah*) remains constitutive of the act of creating and is nourished by the male-female difference. It is in the province of what is 'simply human,' and not just in terms of our genitals, that we can sexually differentiate ourselves" (Falque 2016b, pp. 133–34). Though, one should note that he is not always consistent on this point. In his commentary on Michel Henry's *Incarnation*, Falque suggests that, for Henry, "what instills an anxiety in the dancer in the so-called lovers' night, for example, is, in reality, never the body of his partner as such; for Henry, it is only her flesh or the pathos she experiences as a result of his contact with her hand. However, the body or the sex of the other both attracts and disturbs the man precisely because it is other and different" (Falque 2018, p. 167). Or again, as he remarks in an interview with Richard Kearney, "the man cannot experience what the woman experiences and vice versa because there is a genital difference" (Horton et al. 2019, p. 75).

[9]　As a reviewer helpfully remarked, Falque appears mistaken in his framing of Aristophanes' myth as a contrasting image of the birth of sexual difference, as sexual difference already exists in the differentiation between the three models of Eros (figures which represent heterosexual, male homosexuality, and female homosexuality, respectively). It is not the birth of sexual difference, but rather Eros, that is there illustrated.

[10]　"The erotic imperative by which sexual difference cannot be considered simply to be original, but is also considered originary, part of our destiny, comes not from any author, but from the founder himself, as Christianity understands it: from Christ" (Falque 2016b, p. 139).

[11]　"The difference in unity that makes the Trinity is the same that is repeated in sexualized difference, starting not from a divided unity but from an act of differentiation that is constitutive of the act of loving" (Falque 2016b, p. 141).

[12]　"To say that 'woman came from man, so man comes through woman; but all things come from God' (1 Cor. 11:12) is to make sexual difference originary and willed by God for humanity." Falque (2016b, p. 47). Or again, this sexual difference is "willed by God and by the true vocation of humanity" (Falque 2016b, p. 70).

[13]　Or again, "We thus make up 'one flesh' (Gen. 2:24), as we shall see later, only through the heterogeneity of our different fleshes" (Falque 2016b, p. 115).

[14]　The elision of man/husband and woman/wife is concerning, insofar as it presumes one mode of sexual relation—the nuptial, monogamous, heterosexual relation—to be the normative model of human sexuality. This presumption does not appear intentional, as Falque writes, "the union of flesh is not 'better' in sacramental marriage; it is 'other,' or rather, differently oriented." However, as Richard Kearney has noted, despite this contestation, Falque nevertheless insists that "while lovers may simply be content to be part of humanity—which alone is very significant in their relationship—married spouses search for God, to be incorporated with him and to live their lovemaking in another way." This passage appears to unambiguously elevate the (heterosexual, monogamous) spouse above the mere "lover." (Falque 2016b, p. 170; cited by Kearney in Horton et al. 2019, p. 80).

15　This is not to suggest that he is imprecise on this point. While beyond the scope of the present argument, his methodological text, *Crossing the Rubicon* (Falque 2016a), constitutes an exciting development in contemporary theology and the continental philosophy of religion.

16　On this point, Richard Kearney similarly interrogated Falque on this point, remarking: "your writing on eros seems to me to be overly heterocentric. Heterosexuality seems to be not only normative but mandatory. In fact, at one point you say that sexual difference is "constitutive and originary." Is it heterosexuality that constitutes the modality of sexual difference? Can there not be a sexual difference between same-sex lovers? Does sexual difference have to be biologically gendered and genital? You do say that sexual difference is natural, not cultural—you take on Judith Butler in that regard—but, again, might that not be too exclusive regarding some people? I've just come back from SPEP, and in the conference hotel there were signs saying, "Male, female, and transgender—all welcome." We're not there yet at our respective institutions, but I have the sense that your presuppositions regarding sexual desire are fundamentally heteronormative. In short, my question is: why can homosexual love not also bear witness to the celebration of Eucharistic difference that is the core of your argument? Are not same-sex partners different persons? Different desires? Different bodies, each with his/her own singular uniqueness and thisness (*haecceitas*)?" (Horton et al. 2019, p. 81) Regrettably, Falque chooses not to take up or address this portion of Kearney's questioning.

17　Or again, "the original split is not between the One and the Other, but is strictly inherent to the One; it is the split between the One and its empty place of inscription" (Žižek 2017, p. 72).

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
