# Peer review of "Is There a Homos in Eros: Sexual Incarnation in Emmanuel Falque"

_religions, doi:10.3390/rel14101328_

Round 1
Reviewer 1 Report
- Footnote 8: Italicize "Basic Writings"
- line 249 "they are"
- line 304 "piece"
- Your last critical comments in the "Sexual Difference" section are spot on. What does he actually mean by sexual difference? This is picked up in lines 285-87 in a really nice way.
- Good points in line 370-75: He takes the "I" of gender theory to be the "I that choose myself" to be the nominative and negative "I" of past philosophy, as opposed to the accusative and dative "I".
- 428-430 - Interesting points on the division not between individuals, or individual sexes, but a division in each individual themself. As related to a norm always?
- Your last point is good, but I would like to see it come out a little fuller. Especially as connected to the previous quotes. Perhaps, there is "homos in eros" which means "homos" as any 'other' human being as such, and 'other' subject.
- I would recommend an essay to you (which I was surprised not to see here). It is an interview between Falque and Richard Kearney in the edited volume Somatic Desire; "Embrace and Differentiation: A Phenomenology of Eros". Kearney tells Falque that his work is heterocentric, and that heterosexuality is normative. He asks whether there can be sexual difference between same-sex lovers. Falque never answers the question. At least something to explore and footnote if you want.
- Maybe you can say a bit more on why Falque has a normative view on (hetero)sexuality. Is it because he is a traditional Catholic? Is he thus a theologian before he's a philosopher? Letting his beliefs lead his philosophical points? Not sure how much of this you can or want to cover here. Even a sentence or two as to why you hypothesize he has his position.
Author Response
These are across the board helpful remarks, incorporated into the next draft.
Reviewer 2 Report
The issue that you set out to address is an important one. There are two main areas of where the article would benefit from some rethinking or fuller development. The first has to do with your identification of Levinas as a major source of Falque's heterosexism. The second is the underdevelopment of your argument for your central proposal that there can be a homo in eros.
Regarding Levinas and the feminine: You correctly cite de Beauvoir and Irigaray’s analysis that Levinas relegates women not to the position of the ethical Other, but to a position that amounts to an androcentric objectification and projection. While de Beauvoir and Irigaray are not the only feminist philosophers to read Levinas this way there is a more recent stream of feminist Levinas scholarship that reads his use of figures of the feminine otherwise. On these readings people of any gender could occupy the positions represented in Levinas by feminine or masculine pronouns or figures. This includes the maternal (See especially Guenther), the lover and beloved (see especially Katz and Bergo). I strongly recommend reading their work and, in light of that, reconsidering the role that you give Levinas in your argument about Falque. I suggest that rather than saddling Levinas with heteronormativity, you refer to the heteronormativity that has been read into Levinas, some approvingly (e.g., Dussel, and Falque) and some critically (e.g., de Beauvoir and Irigaray).
Your conclusions about homophobia and heterosexism in Falque seem reasonable. Your references to the relation of this to his theology and biblical hermeneutics makes sense as well. However, this is inconsistent with the specifically Jewish hermeneutic traditions in relation to which Levinas’ references to biblical texts is best understood. While I think it follows from Guenther, Bergo, and Katz’s readings that eros can involve lovers of any gender, Katz (2001) is explicit on this point.
You refer to “the Other” in ways that are likely true of Falque’s usage (and of some others who take some inspiration from Levinas) but which are inconsistent with Levinas’ usage. The ethical Other is not the same as the erotic other (the beloved), nor with others in the political sphere. This can also get confused with the Hegelian other that significantly influences de Beauvoir’s account of woman as other, i.e., “the second sex.” On Levinas’ account the ethical Other occupies the most privileged and commanding position, needless to say, Woman as Other, following de Beauvoir, does not.
At lines 135 and 136 you mention Falque’s view of the “Jewish creation story” in Genesis in comparison to the interpretation by Christ in the Gospels. This reference to the story in Genesis as Jewish is at odds with what would make it Jewish within Jewish hermeneutic traditions. For more on this I recommend Bergo (2018) and ben Pazi.
At 264-5 you mention Falque on the Ish/Isha difference . See Bergo on Talmudic readings of this in Levinas, and especially her citation of Ouaknin, Marc-Alain. The Burnt Book : Reading the Talmud. Princeton, N.J: Princeton University Press, 1998.
At 389-90 you say that Falque “is drawing on a consistent thread, a rhetoric of difference that has, since Levinas, read homosexuality as a narcissistic rejection of alterity.” While many followers of Levinas do identify his critique of a narcissistic, or solipsistic rejection of alterity with a rejection of lived homosexual relations, it is not clear that this was Levinas’ intent. His target is first and foremost eros in Plato. While the most idealized interhuman erotic relations in Plato are between men the object of Levinas’ critique is not the sameness of the sex of beautiful Beloved in interhuman erotic relations. Indeed, according to Diotima’s ladder, eros in its highest form is oriented towards Beauty manifesting in more abstract objects than human persons. Levinas’ critique is aimed at the parthenogenic pregnancy and birth represented by Plato. Pregnancy for Plato is not produced by an interaction between two (necessarily different because they are two not one). One comes into the world pregnant. Birth is inspired by encounters with Beauty, much as what one knows is brought forth by philosophical midwifery. For Levinas, one is changed by one’s encounter with the Other, and the child that is produced is not simply a reiteration of oneself. Both Plato and Levinas use metaphors of sexual reproduction, but we must be careful not to read these metaphors too literally and suppose that they depend on specifically sexual difference. Plato’s use of this imagery clearly does not depend on sexual difference. Levinas can plausibly be read as not depending on it either. His emphasis is on the kind of alterity manifested in the difference between one and the ethical Other, which clearly does not presuppose that either one or the Other is of some particular sex.
Regarding Dussel’s explicit homophobia (especially against lesbian feminism) and his more recent reconsideration if not recantation of this, you might also refer to Alcoff, Linda Martín., and Eduardo. Mendieta. Thinking from the Underside of History : Enrique Dussel’s Philosophy of Liberation. Lanham, Md: Rowman & Littlefield Publishers, 2000, especially Alcoff’s introduction and the reply by Dussel.
The bulk of your argument is spent showing the nature of the homophobia and heterosexism, or at least heteronormativity in Falque’s writings. This part of the argument seems plausible. However, you don’t give very much space to the argument that there could be, or that there is a “homo” in eros. Your references to Bersani and Hocquenghem are promising and perhaps could be more fully explained. There may be more resources in Levinas to support queer eros than can be found in Falque. See Podolsky.
I take your argument to be that non heterosexual lovers have the kind of difference or alterity between them that is essential to eros (and ethics?). That seems right to me. I would suggest that the word homo, in contrast to hetero, obscures that insight. You might substitute queer instead. That is, ask if eros needs alterity, why not queer eros?
Finally, on a minor stylistic point: In the abstract and also in the body of the article you state “Just as the erasure of alterity is understood to render ethics impossible, so too does the erasure of sexual difference render genuine sexuality impossible.” Clearly given the point of the article this is not your view, but the view that you wish to critique. However, the way it is presented is a little ambiguous. It sounds almost as if it were your view. This should be disambiguated to avoid confusion at the start.
Bergo, Bettina, “And God Created Woman” Levinas Studies 12:83-118 (2018).
Guenther, Lisa. The Gift of the Other : Lévinas and the Politics of Reproduction. Albany, NY: State University of New York Press, 2006.
---. “‘Like a Maternal Body’: Emmanuel Levinas and the Motherhood of Moses.” Hypatia 21, no. 1 (2006): 119–136.
Katz, Claire Elise. “‘For Love Is as Strong as Death.’” Philosophy today (Celina) 45 (2001): 124–.
---. Levinas, Judaism, and the Feminine : the Silent Footsteps of Rebecca. Bloomington: Indiana University Press, 2003.
Pazi, Hanoch Ben. “Rebuilding the Feminine in Levinas’s Talmudic Readings.” The Journal of Jewish thought & philosophy 12, no. 3 (2003): 1–32.
Podolsky, Robin. “L’AIMÉ QUI EST L’AIMÉE: Can Levinas’ Beloved Be Queer?” European Judaism 49, no. 2 (2016): 50–70.
Taylor, Chloé. “Lévinasian Ethics and Feminist Ethics of Care.” Symposium (Canadian Society for Continental Philosophy) 9, no. 2 (2005): 217–239.
Author Response
Thank you for your very helpful suggestions. In particular, your suggestion that I check out "Thinking from the Underside of History " was exceptionally useful, and has been incorporated extensively in the conclusion of the article.
Regarding your extensive comments on Levinas, I have taken them into account. I suspect that I still maintain a less generous reading of his phenomenology of eros than you suggest, but I have 1.) softened my language on this point, 2.) have clarified those places where I am referring not to Levinas, but to a particular interpretation of his work (one which I, like you, do not think is a good interpretation of his work), and 3.) have incorporated some of your suggested readings into a footnote that marks alternative feminist interpretations of Levinas' "feminine."
Thank you for your careful reading of this paper, and helpful comments.
Reviewer 3 Report
The author's thesis is correct but could be more forcefully argued.
Falque makes so many mistakes--historical mistakes: for example--his misreading of Aristophanes' myth in the Symposium (if the author is portraying it accurately) should be called out-Aristophanes' figures after the split are 2/3 homo-erotic couples. Sexual difference is not created by Zeus as a punishment--sexual difference already existed before the split, but was not a source of erotic attachment--just a means of reproduction. The story of Aristophanes is about the birth of eros, not the birth of sexual difference. And 2/3 of the story is about women running into women's arms, men running into men's arms.
The author's critique of Falque's " ontology" of sexual difference having no actual content is well-made. But given that critique, the reader is left wondering why the author bothers to engage Falque at all. I think that the author should explain more why Falque is worth reading on this issue or why the author thinks so many people have read Falque without noticing his philosophical errors.
In general, I think this paper has merit but I fail to imagine a thoughtful reader who has read Beauvoir and Butler who doesn't see Falque as philosophical vapid already. I think the author should be more forceful in the argument explaining why Falque's ontology of sexual difference is empirically and phenomenologically inadequate for explaining the lived experience of most people in the 21st century. And the author should add why people might think Falque is important enough to warrant even a critique on this matter.
Author Response
Thank you for your generous remarks.
I added a footnote clarifying that Falque's reading of Aristophanes is questionable.
I have attempted to clarify that I am taking Falque as paradigmatic of a broader (and harmful) approach to alterity, which will hopefully make clear why I find him worth engaging with.
For what it is worth, I don't think his work is vapid, I actually greatly respect his thought and it is really only on this single issue that I have serious problems with him. My hope is to address this concern not in order to delegitimate his work, but to expose a path around this one troubling area.
Thanks again for your helpful comments.
Reviewer 4 Report
The author begins by presenting the position of Emmanuel Falque's understanding of sexuality and the Other. He properly situates the overlap between philosophy and theology that is characteristic of Falque's approach. The concept of analogy is very central to Falque's approach and to understanding of his ontology of sexuality and sexual difference. However, the author's critique of Falque is not founded on an counter-ontology but on political correctness and pleasurability of same-sex or queer relationship. The claim that Falque's ontology of sexual difference is an erasure of "queer love" is at best, dubious. The author needs to counter Falque's position that fecundity and being originary are the fundamental characteristics of sexuality.
Lines 36 and 37: In order to avoid a reductionist understanding of alterity in Levinas, it is essential that the author refers to Totality and Infinity by Levinas.
Lines 59-62: This is a reductive interpretation of Early Heidegger. Heidegger considers the analytic of Dasein in Being and Time to be ontology and not hermeneutics per se. Second, in claiming that language is the house of being, he emphasises that it is through the use of language that we can adequately articulate the meaning of being.
Lines 76-77: To suggest that human finitude refers to sexuality is reductionist. If Falque argues that finitude comprises birth, sexuality and death, then by finitude, as it is in Heideggerian thought, is meant human mortal condition.
Lines 78-80: The quotation does not in any way suggest mere sexuality. Taking on fully the corporal modality, means assuming the human nature in its fullness.
Since Falque holds that the words "eros" and "agape" are neither univocal nor equivocal, it important not forget the analogy that is central in his use of the terms.
Lines 202-204: Does Falque reject biological marker of difference or does he argue the differentiation between man and woman cannot be reduced to just the biological nor just to the social and cultural?
Line 279: Falque's position of not that of mere rejection. He basic argument is that sexual difference is not reductive to the cultural, mythical and biological determinations exclusively.
Lines 298-300: If his pairing is taken in analogous sense, then it would be mistaken to give a simplistic interpretation to it.
Line 304: Altar-piece is incorrectly spelt as "Altar-peice".
Lines 316-321: The critique of Falque is at best erroneous. His use of the "Other" is ontological rooted and is different from "sameness". Secondly sexuality for him is characterised by fecundity and being originary. Hence, eros is used in that sense as it is analogous to agape. Neither homosexuality nor "queer love" accounts for fecundity proper understood. People of the same sex or gender do not exhibit fecundity and so same-sex or queer relationship does not manifest "eros". At best, it depicts "philos".
lines 351-353: This cannot be more generous about sexual diversity. Man/husband and woman/wife are univocal.
Lines 354-356: What the author of the article refers to as "lacuna in Falque's work" is not lacuna. It is a logical consequence of Falque's ontology of sexuality and eros. Homosexuality and any form of "queer love" is not contained within the binary of man/husband and woman/wife that is defended by Falque. A serious objection to Falque should not be based on political correctness or a popular practice. It should be an incisive challenge of Falque's ontology of the Other, sexuality and erotic love. That could be done by introducing a third category that would capture same-sex relationship. Qualifying the the binary by saying "gay man or lesbian woman does not challenge the binary.
Lines 406-408: What exactly is the sameness of homosexuality? Is it sameness of gender, sameness of sex or what? On what ontology is the claimed sameness founded?
Line 420: What the unity of the same is supposed to mean needs to be argued form. How would same-sex relationship to be described if heterosexual relationship is the unity of the same? Is it "unity of the different"?
Line 425-426: Is the person different from the body? There is dualism involved here. More importantly, if it is accepted for the sake of argument that "The queer person is one who lives sexual difference within their own body", it would results in sameness when two queer persons engage in interpersonal love.
Lines 428-430: How does a cut through or within each individual constitute sexual difference?
The quality of English is very good. However minor correction is required in order to correct typographical errors.
Author Response
There were some helpful comments here (e.g. the note on Heidegger) that I have incorporated into the latest draft of the paper. I have also added a footnote clarifying why man/woman is not univocal with husband/wife.
That said, the two assertions that the argument of this paper is dependent on "political correctness" are, to cite the reviewer, "at best, dubious." What I offered here are cogent and concrete arguments against a specific aspect of Falque's phenomenology of eros, not vague gestures to something as ephemeral as political correctness. For that reason, I have not taken those suggestions into consideration in the revision.
Reviewer 5 Report
This is a truly excellent paper and I applaud its author. It can be published without further revisions (which is not something I usually recommend). My only issue with it is that it will be published before my own article on the exact same topic, which I shall now have to revise considerably.
Falque's ignorance of homosexual love, and his consequent inability to think it, is a major problem with his philosophy of love. However, as the author rightly notes, Falque is not at all a reactionary Catholic like Marion who is a very straightforward (and ultimately extremely uninteresting) homophobe. Instead, Falque genuinely has no clue about homosexuality, is not thinking about it at all when he writes about love, and thus falls short on a basic phenomenological level.
Likewise, I have been surprised (and quite frustrated) that none of my colleagues have pointed out this problem with Falque's account, but equally its more vicious instances in Marion and Henry. I am so pleased that someone has finally written this article. (And, I imagine, Emmanuel will be as well.)
I look forward to including this article in the reading list for the course on Falque that I teach and to engage with it in my own work.
Author Response
Thank you for your incredibly generous review.